# Neumann Optimizer: A Practical Optimization Algorithm for Deep Neural Networks

**Shankar Krishnan & Ying Xiao & Rif A. Saurous**
Machine Perception, Google Research
1600 Amphitheatre Parkway
Mountain View, CA 94043, USA
{skrishnan,yingxiao,rif}@google.com

## Abstract

Progress in deep learning is slowed by the days or weeks it takes to train large models. The natural solution of using more hardware is limited by diminishing returns, and leads to inefficient use of additional resources. In this paper, we present a large batch, stochastic optimization algorithm that is both faster than widely used algorithms for fixed amounts of computation, and also scales up substantially better as more computational resources become available. Our algorithm implicitly computes the inverse Hessian of each mini-batch to produce descent directions; we do so without either an explicit approximation to the Hessian or Hessian-vector products. We demonstrate the effectiveness of our algorithm by successfully training large ImageNet models (Inception-V3, Resnet-50, Resnet-101 and Inception-Resnet-V2) with mini-batch sizes of up to 32000 with no loss in validation error relative to current baselines, and no increase in the total number of steps. At smaller mini-batch sizes, our optimizer improves the validation error in these models by 0.8-0.9%. Alternatively, we can trade off this accuracy to reduce the number of training steps needed by roughly 10-30%. Our work is practical and easily usable by others – only one hyperparameter (learning rate) needs tuning, and furthermore, the algorithm is as computationally cheap as the commonly used Adam optimizer.

## 1 Introduction

Large deep neural networks trained on massive data sets have led to major advances in machine learning performance (LeCun et al. (2015)). Current practice is to train networks using gradient descent (SGD) and momentum optimizers, along with natural-gradient-like methods (Hinton et al. (2012); Zeiler (2012); Duchi et al. (2011); Kingma & Ba (2015)). As distributed computation availability increases, total wall-time to train large models has become a substantial bottleneck, and approaches that decrease total wall-time without sacrificing model generalization are very valuable.

In the simplest version of mini-batch SGD, one computes the average gradient of the loss over a small set of examples, and takes a step in the direction of the negative gradient. It is well known that the convergence of the original SGD algorithm (Robbins & Monro (1951)) has two terms, one of which depends on the variance of the gradient estimate. In practice, decreasing the variance by increasing the batch size suffers from diminishing returns, often resulting in speedups that are sublinear in batch size, and even worse, in degraded generalization performance (Keskar et al. (2017)). Some recent work (Goyal et al. (2017); You et al. (2017a;b)) suggests that by carefully tuning learning rates and other hyperparameter schedules, it is possible to train architectures like ResNets and AlexNet on Imagenet with large mini-batches of up to 8192 with no loss of accuracy, shortening training time to hours instead of days or weeks.

There have been many attempts to incorporate second-order Hessian information into stochastic optimizers (see related work below). Such algorithms either explicitly approximate the Hessian (or its inverse), or exploit the use of Hessian-vector products. Unfortunately, the additional computational cost and implementation complexity often outweigh the benefit of improved descent directions. Con-

sequently, their adoption has been limited, and it has largely been unclear whether such algorithms would be successful on large modern machine learning tasks.

In this work, we attack the problem of training with reduced wall-time via a novel stochastic optimization algorithm that uses (limited) second order information without explicit approximations of Hessian matrices or even Hessian-vector products. On each mini-batch, our algorithm computes a descent direction by solving an intermediate optimization problem, and inverting the Hessian of the mini-batch. Explicit computations with Hessian matrices are extremely expensive, so we develop an inner loop iteration that applies the Hessian inverse without explicitly representing the Hessian, or computing a Hessian vector product. The key ingredients in this iteration are the Neumann series expansion for the matrix inverse, and an observation that allows us to replace each occurrence of the Hessian with a single gradient evaluation.

We conduct large-scale experiments using real models (Inception-V3, Resnet-50, Resnet-101, Inception-Resnet-V2) on the ImageNet dataset. Compared to recent work, our algorithm has favourable scaling properties; we are able to obtain linear speedup up to a batch size of 32000, while maintaining or even improving model quality compared to the baseline. Additionally, our algorithm when run using smaller mini-batches is able to improve the validation error by 0.8-0.9% across all the models we try; alternatively, we can maintain baseline model quality and obtain a 10-30% decrease in the number of steps. Our algorithm is easy to use in practice, with the learning rate as the sole hyperparameter.

## 1.1 RELATED WORK

There has been an explosion of research in developing faster stochastic optimization algorithms: there are any number of second-order algorithms that represent and exploit curvature information (Schraudolph et al. (2007); Bordes et al. (2009); Martens & Sutskever (2012); Vinyals & Povey (2012); Sohl-Dickstein et al. (2014); Mokhtari & Ribeiro (2014; 2015); Keskar & Berahas (2016); Byrd et al. (2016); Curtis (2016); Wang et al. (2017a); Martens & Grosse (2015); Grosse & Martens (2016); Bottou et al. (2016)). An alternative line of research has focused on variance reduction (Johnson & Zhang (2013); Shalev-Shwartz & Zhang (2013); Defazio et al. (2014)), where very careful model evaluations are chosen to decrease the variance in the stochastic gradient. Despite the proliferation of new ideas, none of these optimizers have become very popular: the added computational cost and implementation complexity, along with the lack of large-scale experimental evaluations (results are usually reported on small datasets like MNIST or CIFAR-10), have largely failed to convince practitioners that real improvements can be had on large models and datasets. Recent work has focused on using very large batches (Goyal et al. (2017); You et al. (2017a)). These papers rely on careful hyperparameter selection and hardware choices to scale up the mini-batch size to 8192 without degradation in the evaluation metrics.

## 2 ALGORITHMIC IDEAS

Let $x \in \mathbb{R}^d$ be the inputs to a neural net $g(x, w)$ with some weights $w \in \mathbb{R}^n$: we want the neural net to learn to predict a target $y \in \mathbb{R}$ which may be discrete or continuous. We will do so by minimizing the loss function $\mathbb{E}_{(x,y)}\left[\ell(y, g(x, w))\right]$ where $x$ is drawn from the data distribution, and $\ell$ is a per sample loss function. Thus, we want to solve the optimization problem

$$w^* = \operatorname{argmin}_w \mathbb{E}_{(x,y)}\left[\ell(y, g(x, w))\right].$$

If the true data distribution is not known (as is the case in practice), the expected loss is replaced with an empirical loss. Given a set of $N$ training samples $\{(x_1, y_1), (x_2, y_2), \ldots, (x_N, y_N)\}$, let $f_i(w) = \ell(y_i, g(x_i, w))$ be the loss for a particular sample $x_i$. Then the problem we want to solve is

$$w^* = \operatorname{argmin}_w \mathcal{F}(w) = \operatorname{argmin}_w \frac{1}{N} \sum_{i=1}^{N} f_i(w). \tag{1}$$

Consider a regularized first order approximation of $\mathcal{F}(\cdot)$ around the point $w_t$:

$$\mathcal{G}(z) = \mathcal{F}(w_t) + \nabla \mathcal{F}(w_t)^T (z - w_t) + \frac{1}{2\eta} \|z - w_t\|^2$$

Minimizing $\mathcal{G}(\cdot)$ leads to the familiar rule for gradient descent, $w_{t+1} = w_t - \eta \, \nabla \mathcal{F}(w_t)$. If the loss function is convex, we could instead compute a local quadratic approximation of the loss as

$$\mathcal{G}(z) = \mathcal{F}(w_t) + \nabla \mathcal{F}(w_t)^T (z - w_t) + \frac{1}{2}(z - w_t)^T \nabla^2 \mathcal{F}(w_t)(z - w_t), \tag{2}$$

where $\nabla^2 \mathcal{F}(w_t)$ is the (positive definite) Hessian of the empirical loss. Minimizing $\mathcal{G}(z)$ gives the Newton update rule $w_{t+1} = w_t - \left[\nabla^2 \mathcal{F}(w_t)\right]^{-1} \nabla \mathcal{F}(w_t)$. This involves solving a linear system:

$$\nabla^2 \mathcal{F}(w_t)(w - w_t) = -\nabla \mathcal{F}(w_t) \tag{3}$$

Our algorithm works as follows: on each mini-batch, we will form a separate quadratic subproblem as in Equation (2). We will solve these subproblems using an iteration scheme we describe in Section 2.1. Unfortunately, the naive application of this iteration scheme requires a Hessian matrix; we show how to avoid this in Section 2.2. We make this algorithm practical in Section 3.

## 2.1 NEUMANN SERIES

There are many way to solve the linear system in Equation (3). An explicit representation of the Hessian matrix is prohibitively expensive; thus a natural first attempt is to use Hessian-vector products instead. Such a strategy might apply a conjugate gradient or Lanczos type iteration using efficiently computed Hessian-vector products via the Pearlmutter trick (Pearlmutter (1994)) to directly minimize the quadratic form. In our preliminary experiments with this idea, the cost of the Hessian-vector products overwhelms any improvements from a better descent direction (see also Appendix A). We take an even more indirect approach, eschewing even Hessian-vector products.

At the heart of our method lies a power series expansion of the approximate inverse for solving linear systems; this idea is well known, and it appears in various guises as Chebyshev iteration, the conjugate gradient method, the Lanczos algorithm, and Nesterov accelerated gradient methods. In our case, we use the Neumann power series for matrix inverses – given a matrix $A$ whose eigenvalues, $\lambda(A)$ satisfy $0 < \lambda(A) < 1$, the inverse is given by:

$$A^{-1} = \sum_{i=0}^{\infty} (I_n - A)^i.$$

This is the familiar geometric series $(1-r)^{-1} = 1 + r + r^2 + \cdots$ with the substitution $r = (I_n - A)$. Using this, we can solve the linear system $Az = b$ via a recurrence relation

$$z_0 = b \qquad \text{and} \qquad z_{t+1} = (I_n - A)z_t + b, \tag{4}$$

where we can easily show that $z_t \to A^{-1}b$. This is the well known Richardson iteration (Varga (2009)), and is equivalent to gradient descent on the quadratic objective.

## 2.2 QUADRATIC APPROXIMATIONS FOR MINI-BATCHES

A full batch method is impractical for even moderately large networks trained on modest amounts of data. The usual practice is to obtain an unbiased estimate of the loss by using a mini-batch. Given a mini-batch from the training set $(x_{t_1}, y_{t_1}), \ldots, (x_{t_B}, y_{t_B})$ of size $B$, let

$$\hat{f}(w) = \frac{1}{B} \sum_{i=1}^{B} f_{t_i}(w) \tag{5}$$

be the function that we optimize at a particular step. Similar to Equation (2), we form the stochastic quadratic approximation *for the mini-batch* as:

$$\hat{f}(w) \approx \hat{f}(w_t) + \nabla \hat{f}^T (w - w_t) + \frac{1}{2}(w - w_t)^T \left[\nabla^2 \hat{f}\right](w - w_t). \tag{6}$$

As before, we compute a descent direction by solving a linear system, $\left[\nabla^2 \hat{f}\right](w - w_t) = -\nabla \hat{f}$, but now, the linear system is only over the mini-batch. To do so, we use the Neumann series in

Equation (4). Let us assume that the Hessian is positive definite[1], with an operator norm bound $\| \nabla^2 \hat{f} \| < \lambda_{\max}$. Setting $\eta < 1/\lambda_{\max}$, we define the Neumann iterates $m_t$ by making the substitutions $A = \eta \nabla^2 \hat{f}$, $z_t = m_t$, and $b = -\nabla \hat{f}$ into Equation (4):

$$
\begin{aligned}
m_{t+1} &= (I_n - \eta \nabla^2 \hat{f}) m_t - \nabla \hat{f}(w_t) \\
&= m_t - (\boldsymbol{\nabla \hat{f}(w_t) + \eta \nabla^2 \hat{f} m_t}) \\
&\approx m_t - \boldsymbol{\nabla \hat{f}(w_t + \eta m_t)}.
\end{aligned}
\tag{7}
$$

The above sequence of reductions is justified by the following crucial observation: the bold term on the second line is a first order approximation to $\nabla \hat{f}(w_t + \eta m_t)$ for sufficiently small $\|\eta m_t\|$ via the Taylor series:

$$
\nabla \hat{f}(w_t + \eta m_t) = \nabla \hat{f}(w_t) + \eta \nabla^2 \hat{f} m_t + O(\|\eta m_t\|^2).
$$

By using first order only information at a point that is *not* the current weights, we have been able to incorporate curvature information in a matrix-free fashion. This approximation is the sole reason that we pick the slowly converging Neumann series – it allows for extremely cheap incorporation of second-order information. We are now ready to state our idealized Neumann algorithm:

---

**Algorithm 1** Idealized Neumann optimizer

---

**Input:** Initial weights $w_0 \in \mathbb{R}^n$, input data $x_1, x_2, \ldots \in \mathbb{R}^d$, input targets $y_1, y_2, \ldots \in \mathbb{R}$, learning rates $\eta_{in}, \eta_{out}$.

1: **for** $t = 1, 2, 3, \ldots, T$ **do**
2:     Draw a sample $(x_{t_1}, y_{t_1}) \ldots, (x_{t_B}, y_{t_B})$.
3:     Compute derivative: $m_0 = -\nabla \hat{f}(w_t)$.
4:     **for** $k = 1, \ldots, K$ **do**
5:         Update Neumann iterate: $m_k = m_{k-1} - \nabla \hat{f}(w_t + \eta_{in} m_{k-1})$.
6:     Update weights $w_t = w_{t-1} + \eta_{out} m_K$.
7: return $w_T$.

---

The practical solution of Equation (6) occupies the rest of this paper, but let us pause to reflect on what we have done so far. The difference between our technique and the typical stochastic quasi-Newton algorithm is as follows: in an idealized stochastic quasi-Newton algorithm, one hopes to approximate the Hessian of the total loss $\nabla^2 \mathbb{E}_i [f_i(w)]$ and then to invert it to obtain the descent direction $[\nabla^2 \mathbb{E}_i [f_i(w)]]^{-1} \nabla \hat{f}(w)$. We, on the other hand, are content to approximate the Hessian only on the mini-batch to obtain the descent direction $[\nabla^2 \hat{f}]^{-1} \nabla \hat{f}$. These two quantities are fundamentally different, even in expectation, as the presence of the batch in both the Hessian and gradient estimates leads to a product that does not factor. One can think of stochastic quasi-Newton algorithms as trying to find the best descent direction by using second-order information about the total objective, whereas our algorithm tries to find a descent direction by using second-order information implied by the mini-batch. While it is well understood in the literature that trying to use curvature information based on a mini-batch is inadvisable, we justify this by noting that our curvature information arises solely from gradient evaluations, and that in the large batch setting, gradients have much better concentration properties than Hessians.

The two loop structure of Algorithm 1 is a common idea in the literature (for example, Carmon & Duchi (2016); Agarwal et al. (2016); Wang et al. (2017a)): typically though, one solves a difficult convex optimization problem in the inner-loop. In contrast, we solve a much easier linear system in the inner-loop: this idea is also found in (Martens & Sutskever (2012); Vinyals & Povey (2012); Byrd et al. (2016)), where the curvature information is derived from more expensive Hessian-vector products.

Here, we diverge from typical optimization papers for machine learning: instead of deriving a rate of convergence using standard assumptions on smoothness and strong convexity, we move onto the much more poorly defined problem of building an optimizer that actually works for large-scale deep neural nets.

---

[1]We will show how to remove the positive definite assumption in Section 3.1

## 3  AN OPTIMIZER FOR NEURAL NETWORKS

Our idealized Neumann optimizer algorithm is deeply impractical. The main problems are:

1. We assumed that the expected Hessian is positive definite, and furthermore that the Hessian on each mini-batch is also positive definite.

2. There are four hyperparameters that significantly affect optimization – $\eta_{in}, \eta_{out}$, inner loop iterations and batch size.

We shall introduce two separate techniques for convexifying the problem – one for the total Hessian and one for mini-batch Hessians, and we will reduce the number of hyperparameters to just a single learning rate.

### 3.1  CONVEXIFICATION

In a deterministic setting, one of the best known techniques for dealing with non-convexity in the objective is cubic regularization (Nesterov & Polyak (2006)): adding a regularization term of $\frac{\alpha}{3} \|w - w_t\|^3$ to the objective function, where $\alpha$ is a scalar hyperparameter weight. This is studied in Carmon & Duchi (2016), where it is shown that under mild assumptions, gradient descent on the regularized objective converges to a second-order stationary point (i.e., Theorem 3.1). The cubic regularization method falls under a broad class of trust region methods. This term is essential to theoretically guarantee convergence to a critical point. We draw inspiration from this work and add two regularization terms – a cubic regularizer, $\frac{\alpha}{3} \|w - v_t\|^3$, and a repulsive regularizer, $\beta / \|w - v_t\|$ to the objective, where $v_t$ is an exponential moving average of the parameters over the course of optimization. The two terms oppose each other - the cubic term is attractive and prevents large updates to the parameters especially when the learning rate is high (in the initial part of the training), while the second term adds a repulsive potential and starts dominating when the learning rate becomes small (at the end of training). The regularized objective is $\hat{g}(w) = \hat{f}(w) + \frac{\alpha}{3} \|w - v_t\|^3 + \beta / \|w - v_t\|$ and its gradient is

$$\nabla \hat{g}(w) = \nabla \hat{f}(w) + \left( \alpha \|w - v_t\|^2 - \frac{\beta}{\|w - v_t\|^2} \right) \frac{w - v_t}{\|w - v_t\|} \tag{8}$$

Even if the expected Hessian is positive definite, this does not imply that the Hessians of individual batches themselves are also positive definite. This poses substantial difficulties since the intermediate quadratic forms become unbounded, and have an arbitrary minimum in the span of the subspace of negative eigenvalues. Suppose that the eigenvalues of the Hessian, $\lambda(\nabla^2 \hat{g})$, satisfy $\lambda_{\min} < \lambda(\nabla^2 \hat{g}) < \lambda_{\max}$, then define the coefficients:

$$\mu = \frac{\lambda_{\max}}{|\lambda_{\min}| + \lambda_{\max}} \qquad \text{and} \qquad \eta = \frac{1}{\lambda_{\max}}.$$

In this case, the matrix $\hat{B} = (1 - \mu)I_n + \mu\eta\nabla^2 \hat{g}$ is a positive definite matrix. If we use this matrix instead of $\nabla^2 \hat{f}$ in the inner loop, we obtain updates to the descent direction:

$$
\begin{aligned}
m_k &= (I_n - \eta\hat{B})m_{k-1} - \nabla\hat{g}(w_t) \\
&= \left( \mu I_n - \mu\eta\nabla^2\hat{g}(w_t) \right) m_{k-1} - \nabla\hat{g}(w_t) \\
&= \mu m_{k-1} - \left( \nabla\hat{g}(w_t) + \eta\mu\nabla^2\hat{g}(w_t)m_{k-1} \right) \\
&\approx \mu m_{k-1} - \eta\nabla\hat{g}(w_t + \mu m_{k-1}).
\end{aligned}
\tag{9}
$$

It is not clear *a priori* that the matrix $\hat{B}$ will yield good descent directions, but if $|\lambda_{\min}|$ is small compared to $\lambda_{\max}$, then the perturbation does not affect the Hessian beyond a simple scaling. This is the case later in training (Sagun et al. (2016); Chaudhari et al. (2016); Dauphin et al. (2014)), but to validate it, we conducted an experiment (see Appendix A), where we compute the extremal mini-batch Hessian eigenvalues using the Lanczos algorithm. Over the trajectory of training, the following qualitative behaviour emerges:

- Initially, there are many large negative eigenvalues.

Table 1: Summary of Hyperparameters.

| Hyperparameter | Setting |
|---|---|
| Cubic Regularizer | $\alpha = 10^{-7}$ |
| Repulsive Regularizer | $\beta = 10^{-5} \times$ num variables |
| Moving Average | $\gamma = 0.99$ |
| Momentum | $\mu \propto \left(1 - \frac{1}{1+t}\right)$, starting at $\mu = 0.5$ and peaking at $\mu = 0.9$. |
| Number of SGD warm-up steps | num SGD steps $= 5$ epochs |
| Number of reset steps | K, starts at 10 epochs, and doubles after every reset. |

- During the course of optimization, these large negative eigenvalues decrease in magnitude towards zero.

- Simultaneously, the largest positive eigenvalues continuously increase (almost linearly) over the course of optimization.

This validates our mini-batch convexification routine. In principle, the cubic regularizer is redundant – if each mini-batch is convex, then the overall problem is also convex. But since we only crudely estimate $\lambda_{\min}$ and $\lambda_{\max}$, the cubic regularizer ensures convexity without excessively large distortions to the Hessian in $\hat{B}$. Based on the findings in our experimental study, we set $\mu \propto 1 - \frac{1}{1+t}$, and $\eta \propto 1/t$.

## 3.2 RUNNING THE OPTIMIZER: SGD BURN IN AND INNER LOOP ITERATIONS

We now make some adjustments to the idealized Neumann algorithm to improve performance and stability in training. The first change is trivial – we add a very short phase of vanilla SGD at the start. SGD is typically more robust to the pathologies of initialization than other optimization algorithms, and a "warm-up" phase is not uncommon (even in a momentum method, the initial steps are dampened by virtue of using exponential weighted averages starting at 0).

Next, there is an open question of how many inner loop iterations to take. Our experience is that there are substantial diminishing marginal returns to reusing a mini-batch. A deep net has on the order of millions of parameters, and even the largest mini-batch size is less than fifty thousand examples. Thus, we can not hope to rely on very fine-grained information from each mini-batch. From an efficiency perspective, we need to keep the number of inner loop iterations very low; on the other hand, this leads to the algorithm degenerating into an SGD-esque iteration, where the inner loop descent directions $m_t$ are never truly useful. We solve this problem as follows: instead of freezing a mini-batch and then computing gradients with respect to this mini-batch at every iteration of the inner loop, we compute a stochastic gradient at every iteration of the inner loop. One can think of this as solving a stochastic optimization subproblem in the inner loop instead of solving a deterministic optimization problem. This small change is effective in practice, and also frees us from having to carefully pick the number of inner loop iterations – instead of having to carefully balance considerations of optimization quality in the inner loop with overfitting on a particular mini-batch, the optimizer now becomes relatively insensitive to the number of inner loop iterations; we pick a doubling schedule for our experiments, but a linear one (as presented in Algorithm 2) works equally well. Additionally, since the inner and outer loop updates are now identical, we simply apply a single learning rate $\eta$ instead of two.

Finally, there is the question of how to set the mini-batch size for our algorithm. Since we are trying to extract second-order information from the mini-batch, we hypothesize that Neumann optimizer is better suited to the large batch setting, and that one should pick the mini-batch size as large as possible. We provide experimental evidence for this hypothesis in Section 4.

As an implementation simplification, the $w_t$ maintained in Algorithm 2 are actually the displaced parameters $(w_t + \mu m_t)$ in Equation (7). This slight notational shift then allows us to "flatten" the two loop structure with no change in the underlying iteration. In Table 1, we compile a list of hyperparameters that work across a wide range of models (all our experiments, on both large and small models, used these values): the only one that the user has to select is the learning rate.

---

**Algorithm 2** Neumann optimizer: Learning rate $\eta(t)$, cubic regularizer $\alpha$, repulsive regularizer $\beta$, momentum $\mu(t)$, moving average parameter $\gamma$, inner loop iterations $K$

---

**Input:** Initial weights $w_0 \in \mathbb{R}^n$, input data $x_1, x_2, \ldots \in \mathbb{R}^d$, input targets $y_1, y_2, \ldots \in \mathbb{R}$.
1: Initialize moving average weights $v_0 = w_0$ and momentum weights $m_0 = \mathbf{0}$.
2: Run vanilla SGD for a small number of iterations.
3: **for** $t = 1, 2, 3, \ldots, T$ **do**
4:     Draw a sample $(x_{t_1}, y_{t_1}), \ldots, (x_{t_B}, y_{t_B})$.
5:     Compute derivative $\nabla \hat{f} = (1/B) \sum_{i=1}^{B} \nabla \ell(y_{t_i}, g(x_{t_i}, w_t))$
6:     **if** $t = 1$ modulo $K$ **then**
7:         Reset Neumann iterate $m_t = -\eta \nabla \hat{f}$
8:     **else**
9:         Compute update $d_t = \nabla \hat{f} + \left( \alpha \|w_t - v_t\|^2 - \frac{\beta}{\|w_t - v_t\|^2} \right) \frac{w_t - v_t}{\|w_t - v_t\|}$
10:       Update Neumann iterate: $m_t = \mu(t)m_{t-1} - \eta(t)d_t$.
11:       Update weights: $w_t = w_{t-1} + \mu(t)m_t - \eta(t)d_t$.
12:       Update moving average of weights: $v_t = w_t + \gamma(v_{t-1} - w_t)$
13: return $w_T - \mu(T)m_T$.

---

## 4 EXPERIMENTS

We experimentally evaluated our optimizer on several large convolutional neural networks for image classification[2]. While our experiments were successful on smaller datasets (CIFAR-10 and CIFAR-100) without any hyperparameter modifications, we shall only report results on the ImageNet dataset.

Our experiments were run in Tensorflow (Abadi et al.), on Tesla P100 GPUs, in our distributed infrastructure. To abstract away the variability inherent in a distributed system such as network traffic, job loads, pre-emptions etc, we use training epochs as our notion of time. Since we use the same amount of computation and memory as an Adam optimizer (Kingma & Ba (2015)), our step times are on par with commonly used optimizers. We used the standard Inception data augmentation (Github (2017)) for all models. We used an input image size of $299 \times 299$ for the Inception-V3 and Inception-Resnet-V2 models, and $224 \times 224$ for all Resnet models, and measured the evaluation metrics using a single crop. We intend to open source our code at a later date.

Neumann optimizer seems to be robust to different initializations and trajectories (see Appendix ). In particular, the final evaluation metrics are stable do not vary significantly from run to run, so we present results from single runs throughout our experimental section.

### 4.1 FIXED MINI-BATCH SIZE: BETTER ACCURACY OR FASTER TRAINING

First, we compared our Neumann optimizer to standard optimization algorithms fixing the mini-batch size. To this end, for the baselines we trained an Inception-V3 model (Szegedy et al. (2016)), a Resnet-50 and Resnet-101 (He et al. (2016a;b)), and finally an Inception-Resnet-V2 (Szegedy et al. (2017)). The Inception-V3 and Inception-Resnet-V2 models were trained as in their respective papers, using the RMSProp optimizer (Hinton et al. (2012)) in a synchronous fashion, additionally increasing the mini-batch size to 64 (from 32) to account for modern hardware. The Resnet-50 and Resnet-101 models were trained with a mini-batch size of 32 in an asynchronous fashion using SGD with momentum 0.9, and a learning rate of 0.045 that decayed every 2 epochs by a factor of 0.94. In all cases, we used 50 GPUs. When training synchronously, we scale the learning rate linearly after an initial burn-in period of 5 epochs where we slowly ramp up the learning rate as suggested by Goyal et al. (2017), and decay every 40 epochs by a factor of 0.3 (this is a similar schedule to the asynchronous setting because $0.94^{20} \approx 0.3$). Additionally, we run Adam to compare against a popular baseline algorithm.

We evaluate our optimizer in terms of final test accuracy (top-1 validation error), and the number of epochs needed to achieve a fixed accuracy. In Figure 2, we can see the training curves and the

---

[2]Although deep nets are our primary concern, we study the performance on a stochastic convex optimization problem in Appendix B.

Table 2: Final Top-1 Validation Error

|  | Baseline | Neumann | Improvement |
|---|---|---|---|
| Inception-V3 | 21.7 % | 20.8 % | **0.91%** |
| Resnet-50 | 23.9 % | 23.0 % | **0.94 %** |
| Resnet-101 | 22.6 % | 21.7% | **0.86 %** |
| Inception-Resnet-V2 | 20.3 % | 19.5 % | **0.84 %** |

test error for Inception V3 as compared to the baseline RMSProp. The salient characteristics are

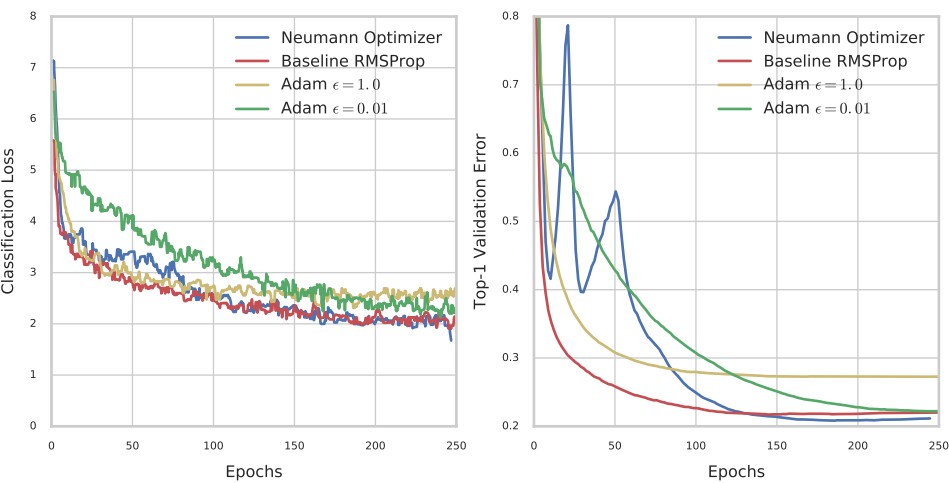

Figure 1: Training and Evaluation curves for Inception V3.

as follows: first, the classification loss (the sum of the main cross entropy loss and the auxiliary head loss) is *not* improved, and secondly there are oscillations early in training that also manifest in the evaluation. The oscillations are rather disturbing, and we hypothesize that they stem from slight mis-specification of the hyperparameter $\mu$, but all the models we train appear to be robust to these oscillations. The lack of improvement in classification loss is interesting, especially since the evaluation error is improved by a non-trivial increment of 0.8-0.9 %. This improvement is consistent across all our models (see Table 2 and Figure 2). As far as we know, it is unusual to obtain an improvement of this quality when changing from a well-tuned optimizer. We discuss the open problems raised by these results in the discussion section.

This improvement in generalization can also traded-off for faster training: if one is content to obtain the previous baseline validation error, then one can simply run the Neumann optimizer for fewer steps. This yields a 10-30% speedup whilst maintaining the current baseline accuracy.

On these large scale image classification models, Adam shows inferior performance compared to both Neumann optimizer and RMSProp. This reflects our understanding that architectures and algorithms are tuned to each other for optimal performance. For the rest of this paper, we will compare Neumann optimizer with RMSProp only.

## 4.2 LINEAR-SCALING AT VERY LARGE BATCH SIZES

Earlier, we hypothesized that our method is able to efficiently use large batches. We study this by training a Resnet-50 on increasingly large batches (using the same learning rate schedule as in Section 4.1) as shown in Figure 3 and Table 3. Each GPU can handle a mini-batch of 32 examples, so for example, a batch size of 8000 implies 250 GPUs. For batch sizes of 16000 and 32000, we used 250 GPUs, each evaluating the model and its gradient multiple times before applying any updates. Our algorithm scales to very large mini-batches: up to mini-batches of size 32000, we are still better than the baseline. To our knowledge, our Neumann Optimizer is a new state-of-the-art in taking advantage of large mini-batch sizes while maintaining model quality. Compared to Goyal

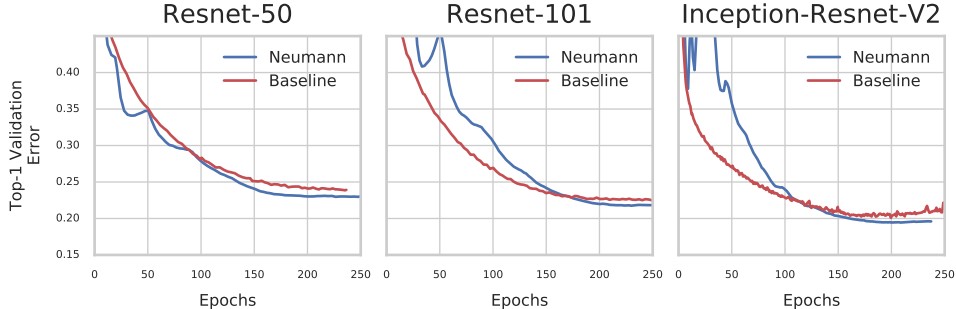

Figure 2: Comparison of Neumann optimizer with hand-tuned optimizer on different ImageNet models.

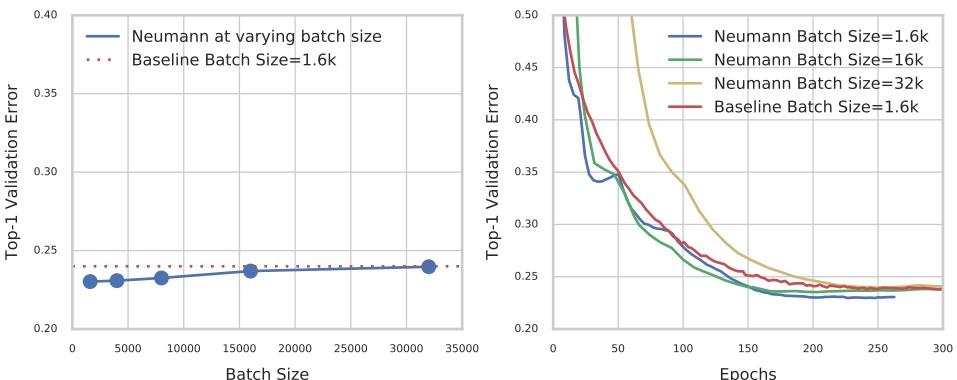

Figure 3: Scaling properties of Neumann optimizer vs SGD with momentum.

et al. (2017), it can take advantage of 4x larger mini-batches; compared to You et al. (2017b;a) it uses the same mini-batch size but matches baseline accuracy while You et al. (2017b;a) suffers from a 0.4-0.7% degradation.

Table 3: Scaling Performance of our Optimizer on Resnet-50

| Batch Size | # workers | Top-1 Validation Error | # Epochs | Param. updates |
|---|---|---|---|---|
| 1600 | 50 | 23.0 % | 226 | 181K |
| 4000 | 125 | 23.0 % | 230 | 73.6K |
| 8000 | 250 | 23.1 % | 258 | 41.3K |
| 16000 | 500 | 23.5 % | 210 | 16.8K |
| 32000 | 1000 | 24.0 % | 237 | 9.5K |

## 4.3 EFFECT OF REGULARIZATION

We studied the effect of regularization by performing an ablation experiment (setting $\alpha$ and $\beta$ to 0). Our main findings are summarized in Table 4 (and Figure 6 in Appendix C). We can see that regularization improves validation performance, but even without it, there is a performance improvement from just running the Neumann optimizer.

Table 4: Effect of regularization - Resnet-50, batch size 4000

| Method | Top-1 Error |
|---|---|
| Baseline | 24.3 % |
| Neumann (without regularization) | 23.5 % |
| Neumann (with regularization) | 23.0 % |

### 4.4 NEGATIVE RESULT FOR SEQUENCE-TO-SEQUENCE RNNS

We also tried our algorithm on a large-scale sequence-to-sequence speech-synthesis model called Tacotron (Wang et al. (2017b)), where we were unable to obtain any speedup or quality improvements. Training this model requires aggressive gradient clipping; we suspect the Neumann optimizer responds poorly to this, as our approximation of the Hessian in Equation (7) breaks down.

## 5 DISCUSSION

In this paper, we have presented a large batch optimization algorithm for training deep neural nets; roughly speaking, our algorithm implicitly inverts the Hessian of individual mini-batches. Our algorithm is practical, and the only hyperparameter that needs tuning is the learning rate. Experimentally, we have shown the optimizer is able to handle very large mini-batch sizes up to 32000 without any degradation in quality relative to current baseline models. Intriguingly, at smaller mini-batch sizes, the optimizer is able to produce models that generalize better, and improve top-1 validation error by 0.8-0.9% across a variety of architectures with no attendant drop in the classification loss.

We believe the latter phenomenon is worth further investigation, especially since the Neumann optimizer does not improve the training loss. This indicates that, somehow, the optimizer has found a different local optimum. We think that this confirms the general idea that optimization and generalization can not be decoupled in deep neural nets.

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

# A  Mini-batch Krylov algorithms, and Hessian Eigenvalue Estimates

There are many possible strategies for solving the quadratic mini-batch optimization problem. In particular, various Krylov subspace methods (Martens & Sutskever (2012); Vinyals & Povey (2012)), such as conjugate gradient, are very appealing because of their fast convergence and ability to solve the linear system in Equation (3) using Hessian-vector products. Unfortunately, in our preliminary experiments, none of these Krylov methods gave better or faster optimizers than SGD (and its variants) – the Hessian-vector product was simply too expensive relative to the quality of the descent directions.

On the other hand, the closely related idea of running a Lanczos algorithm on the mini-batch gives us excellent information about the eigenvalues of the mini-batch Hessian. The Lanczos algorithm is a Krylov subspace method for computing the eigenvalues of a Hermitian matrix (see Trefethen & Bau (1997) for a detailed exposition). After $k$ iterations, the Lanczos algorithm outputs a $k \times k$ tridiagonal matrix whose eigenvalues, known as Ritz values, typically are close to the extreme (largest magnitude) eigenvalues of the original matrix. Crucially, the Lanczos algorithm requires only the ability to perform matrix-vector products; in our setting, one can compute Hessian-vector products almost as cheaply as the gradient using the Pearlmutter trick (Pearlmutter (1994)), and thus we can use the Lanczos algorithm to compute estimates of the extremal eigenvalues of the Hessian.

Supposing that we have an upper bound on the most positive eigenvalue $\lambda_{\max}$, then by applying a shift operation to the Hessian of the form $\nabla^2 \hat{f} - \lambda_{\max} I_n$, we can compute the most negative eigenvalue $\lambda_{\min}$. This is useful when $|\lambda_{\min}| \ll |\lambda_{\max}|$ for example.

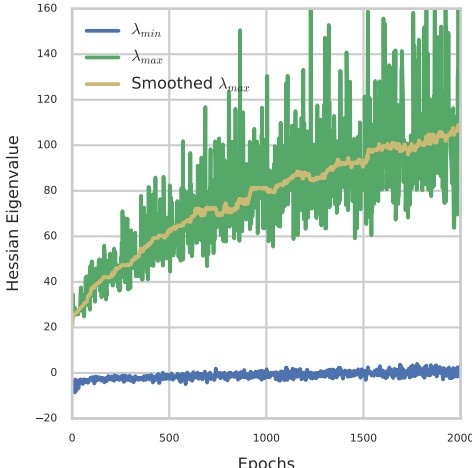

Figure 4: Minimum and Maximum Eigenvalues of a CIFAR-10 model. Batch size=128, using 10 iterations of the Lanczos algorithm.

The following is an experiment that we ran on a CIFAR-10 model: we trained the model as per usual using SGD. Along the trajectory of optimization, we ran a Lanczos algorithm to estimate the most positive and most negative eigenvalues. Figure 4 depicts these eigenvalues. Although the estimates of the mini-batch eigenvalues are very noisy, the qualitative behaviour is still clear:

- The maximum eigenvalue increases (almost) linearly over the course of optimization.
- The most negative eigenvalue decays towards 0 (from below) over the course of optimization.

This is consistent with the existing results in the literature (Sagun et al. (2016); Chaudhari et al. (2016); Dauphin et al. (2014)), and we use these observations to specify a parametric form for the $\mu$ parameter.

## B  PERFORMANCE ON CONVEX PROBLEMS

In this section, we will compare the performance of the Neumann optimizer with other stochastic optimizers on a convex problems. We generated a synthetic binary classification problem – the problem is to learn a linear classifier over points sampled from a Gaussian distribution via logistic regression. The input features were 100-dimensional vectors and the condition number of the Hessian was roughly $10^4$ (since it changes over the course of the optimization). We used a small weight decay of $10^{-6}$ – without weight decay, the problem is ill-posed.

We compared the performance of SGD, Adam, and Neumann optimizers on the problem for batch sizes of 10 and 200 (with learning rate 0.05 and 0.5). Since the original and stochastic problems are convex, $\alpha$ and $\beta$ are set to 0 for the Neumann optimizer. Additionally, we studied a true second order Newton algorithm: with a little hyperparameter tuning, we set the learning rate higher by a factor of 20, and in addition, we allowed the Newton algorithm special access to Hessian estimates from a separate mini-batch of 500 samples.

In Figure 5, we plot the training loss. The major observations are:

1. There is almost no difference in the performance of SGD in comparison with Adam.
2. Neumann does considerably better than SGD and Adam in getting the cost down.
3. The Newton algorithm is better than Neumann optimizer at larger batch sizes (though we have not accounted for neither the additional samples needed to estimate the Hessian nor the substantial computational cost of inverting a full Hessian).

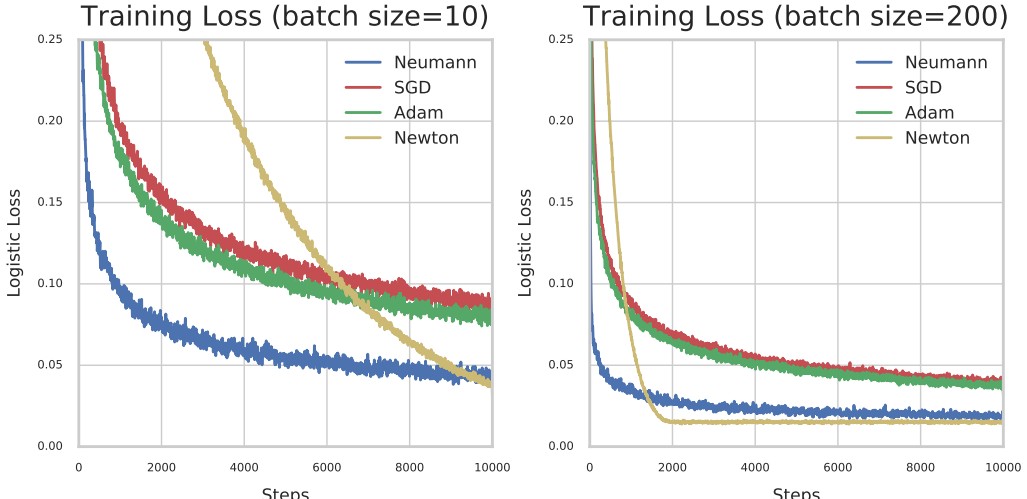

Figure 5: Comparison between SGD, Adam, Newton and Neumann optimizers on a synthetic logistic regression problem (a) with batch size 10, and (b) with batch size 200.

## C  EFFECT OF REGULARIZATION

In this section, we study the effects of removing the cubic and repulsive regularizer terms in the objective. In Figure 6, the output models are of lower quality, though the final evaluation metrics are still better than a baseline RMSProp.

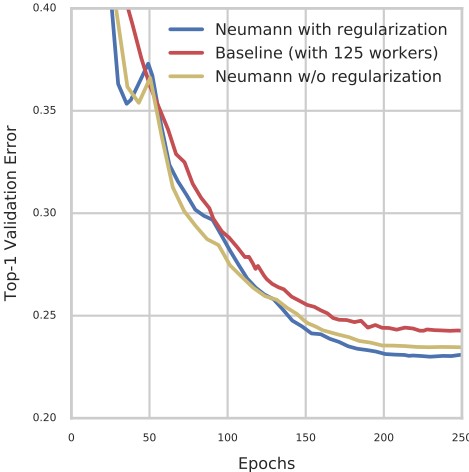

Figure 6: Effect of regularization on Resnet-50 architecture. Note that we used 125 GPUs (minibatch size of 4000) in this experiment.

## D  ROBUSTNESS TO INITIALIZATIONS AND RANDOMNESS

In this section, we compare four different initializations and trajectories of the Neumann optimizer. In Figure 7, although the intermediate training losses and evaluation metrics are different, the final output model quality is the same, and are substantially better than RMSProp.

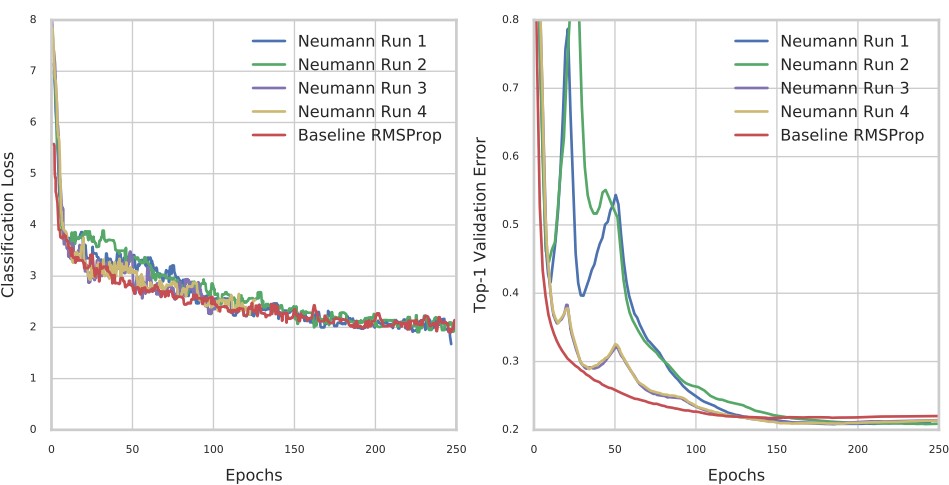

Figure 7: Multiple runs of Neumann optimizer on Inception-V3. This is a 50 GPU experiment.

