# OpenReview forum: "Neumann Optimizer: A Practical Optimization Algorithm for Deep Neural Networks"
_ICLR.cc/2018/Conference — Accept (Poster)_

### Official Review · AnonReviewer3 · 2017-11-24
**See below.**

**Rating:** 6
**Confidence:** 3

**Review:**


This paper presents a new 2nd-order algorithm that implicitly uses curvature information, and it shows the intuition behind the approximation schemes in the algorithms and also validates the heuristics in various experiments.  The method involves using Neumann Series and Richardson iteration to avoid Hessian-vector product in second order method for NN.  In the actual performance, the paper presents both practical efficiency and better generalization error in different deep neural networks for image classification tasks, and the authors also show differences according to different settings, e.g., Batch Size, Regularization.  The numerical examples are relatively clear and easy to figure out details.

1. While the paper presents the algorithm as an optimization algorithm, although it gets better learning performance, it would be interesting to see how well it is as an optimizer.  For example, one simple experiment would be showing how it works for convex problems, e.g., logistic regression.  Realistic DNN systems are very complex, and evaluating the method in a simple setting would help a lot in determining what if anything is novel about the method.

2. Also, for deep learning problems, it would be more convincing to see how different initialization can affect the performances.

3. Although the authors present their algorithm as a second order method at beginning, the final algorithm is kind of like a complex momentum SGD with limited memory.  Rather than simply throwing out a new method with a new name, it would be helpful to understand what the steps of this method are implicitly doing.  Please explain more about this.

4. It said that the algorithm is hyperparameter free except for learning rate.  However, it is hard to see why there is no need to tune other hyperparameters, e.g., Cubic Regularizer, Repulsive Regularizer.  The effect/sensitivity of hyperparameters for second order methods are quite different than hyperparameters for first order methods, and it is of interest to know how hyperparameters for implicit second order methods perform.

5. For Section 4.2, the well know benefit by using large batch size to train models is that it could reduce training time and epochs.  However, from Table 3, there is no such phenomenon.  Please explain.

---

> ### Author Response · Authors · 2018-01-05
> **Combined response to AnonReviewer3's comments**
>
> Thank you AnonReviewer3 for your thoughts and comments: we address your comments below and hope to clear up one misconception (caused by poor labelling of Table 3):
>
> 1. We have added an experiment in Appendix B to show the results on a synthetic logistic regression problem. We compared the Neumann optimizer with SGD, Adam and a Newton algorithm for varying batch sizes. Our method outperforms SGD and Adam consistently, and while Newton’s method descends to a better loss, it comes at a steep per-step cost. We believe there are other large batch methods like Nesterov and SVRG that might get to lower losses than our method. However, none of these algorithms perform well on training a deep neural net.
>
> 2. We've included an Appendix D with a new experiment illustrating that different initializations and trajectories of optimization all give the same quality model output (for the Inception V3 model).
>
> 3. We're not quite sure what the reviewer is looking for here: it seems that Section 2 gives a derivation of the method: the method is implicitly inverting the Hessian (which is convexified after regularization) of a mini-batch. Our algorithm crucially differs from standard momentum in that gradient evaluation occurs at a different point from the current iterate (in Algorithm 1), and we are not applying an exponential decay (a standard momentum update would blow up if you did this).
>
> 4. We agree that it is of interest to further study the sensitivity to hyperparameters. The results that we have hold for ImageNet, but also for CIFAR-10 and CIFAR-100 with no change in hyperparameters, so we think that the results are likely to carry over to most modern CNN architectures on image datasets -- the hyperparameter choice will likely work out of the box (much like the beta_1, beta_2 and epsilon parameters in Adam). We agree that there appears to be quite a few hyperparameters, but \alpha and \beta are regularization coefficients, so they have to be roughly scaled to the loss; \gamma is a moving average coefficient and never needs to be changed; \mu is dependent only on time, not the model; finally, training is quite insensitive to K (as mentioned in Section 3.2). Thus, the only hyperparameter that needs to be specified is the learning rate \eta, and that does determine the speed of optimization.
>
> 5. The epochs listed in Table 3 are total epochs (i.e., total sum of all samples seen by all workers), so using twice as many workers is in fact twice as fast (we've updated the table to clarify this). We're a little concerned that we were not clear on the significance of the experimental results: our algorithm scales up to a batch size of 32000 (beating state-of-the-art for large batch training), and we obtain linear speedups across this regime i.e., we can run 500 workers, in 1/10th the time that it takes the usual 50 worker baseline. We think of this as the major contribution of our work.

---

### Official Review · AnonReviewer1 · 2017-11-25
**Neumann Optimizer: A Practical Optimization Algorithm for Deep Neural Networks**

**Rating:** 6
**Confidence:** 3

**Review:**

Summary:
The paper proposes Neumman optimizer, which makes some adjustments to the idealized Neumman algorithm to improve performance and stability in training. The paper also provides the effectiveness of the algorithm by training ImageNet models (Inception-V3, Resnet-50, Resnet-101, and Inception-Resnet-V2).

Comments:
I really appreciate the author(s) by providing experiments using real models on the ImageNet dataset. The algorithm seems to be easily used in practice.

I do not have many comments for this paper since it focuses only in practical view without theory guarantee rigorously.

As you mention in the paper that the algorithm uses the same amount of computation and memory as Adam optimizer, but could you please provide the reason why you only compare Neumann Optimizer with Baseline RMSProp but not with Adam? As we know, Adam is currently very well-known algorithm to train DNN. Do you think it would be interesting if you could compare the efficiency of Neumann optimizer with Adam? I understand that you are trying to improve the existing results with their optimizer, but this paper also introduces new algorithm.

The question is that, with the given architectures and dataset, what algorithm should people consider to use between Neumann optimizer and Adam? Why should people use Neumann optimizer but not Adam, which is already very well-known? If Neumann optimizer can surpass Adam on ImageNet, I think your algorithm will be widely used after being published.

Minor comments:
Page 3, in eq. (3): missing “-“ sign
Page 3, in eq. (6): missing “transpose” on \nabla \hat{f}
Page 4, first equation: O(|| \eta*mu_t ||^2)
Page 5, in eq. (9): m_{k-1}

---

> ### Author Response · Authors · 2018-01-05
> **Combined response to AnonReviewer1's comments**
>
> Thank you AnonReviewer1 for your feedback and comments.
>
> We ran a new set of experiments comparing Adam, RMSprop and Neumann (Figure 1). Adam achieves similar (or worse) results to the RMSprop baselines: in comparison to our Neumann optimizer, the training is slower, the output model is lower quality, and the optimizer scales poorly. When training with Adam, we observed instability with default parameters (especially, epsilon). We changed it to 0.01 and 1.0 and have two runs which show dramatically different results. Our initial reason for not including comparisons to Adam was that we wanted to use standard models and training parameters (i.e., the Inception and Resnet papers use RMSprop).
>
> We hope that practitioners will consider Neumann over Adam for the following reasons:
> - Significantly higher quality output models when training using few GPUs.
> - Ability to scale up to vastly more GPUs/TPUs, and overall decreased training time.
>
> We’ve incorporated your minor comments -- thanks again!

---

> > ### Comment · AnonReviewer1 · 2018-01-05
> > **Neumann Optimizer**
> >
> > Thank you for adding more experiments.
> >
> > In my opinion, it is hard to judge your paper since you do not have any theoretical guarantee rigorously. But it seems that your algorithm is promising. Therefore, I increased the rating score for giving it a chance to be published. I hope the practitioners will try to use it and see if there is any drawback.

---

### Official Review · AnonReviewer2 · 2017-11-27
**[UPDATED] Interesting idea, nice experiments, good motivation but lacks theoretical understanding**

**Rating:** 6
**Confidence:** 4

**Review:**

The paper proposes a new algorithm, where they claim to use Hessian implicitly and are using a motivation from power-series. In general, I like the paper.

To me, Algorithm 1 looks like some kind of proximal-point type algorithm. Algorithm 2 is more heuristic approach, with a couple of parameters to tune it.  Given the fact that there is convergence analysis or similar theoretical results, I would expect to have much more numerical experiments. E.g. there is no results of Algorithm 1. I know it serves as a motivation, but it would be nice to see how it works.

Otherwise, the paper is clearly written.
The topic is important, but I am a bit afraid of significance. One thing what I do not understand is, that why they did not compare with Adam? (they mention Adam algorithm soo many times, that it should be compared to).

I am also not sure, how sensitive the results are for different datasets? Algorithm 2 really needs so many parameters (not just learning rate). How \alpha, \beta, \gamma, \mu, \eta, K influence the speed? how sensitive is the algorithm for different choices of those parameters?

---

> ### Author Response · Authors · 2018-01-05
> **Combined response to AnonReviewer2's comments**
>
> Thank you AnonReviewer2 for your comments. Here are our responses:
>
> We have added a number of new experiments, including (1) Solving a stochastic convex optimization problem (where the Neumann optimizer is far better than SGD or Adam), (2) Comparisons with Adam on Inception-V3 (see below) and (3) Multiple runs of the Neumann algorithm on Inception-V3 showing that the previous experiments are reproducible.
>
> To the comment about running Algorithm 1: we’ve run it on stochastic convex problems before, where it performs much better than either SGD or Adam. On deep neural nets, our earlier experience with similar “two-loop” algorithms (i.e., freeze the mini-batch, and perform substantial inner-loop computation) lead us to the conclusion that Algorithm 1 would most likely not perform very well at training deep neural nets. The main difficulty is that the inner loop iterations “overfit” to the mini-batch. As you mentioned, this is meant to purely motivational for Algorithm 2.
>
> Adam achieves similar (or worse) results to the RMSprop baselines (Figure 1): in comparison to our Neumann optimizer, the training is slower, the output model is lower quality, and the optimizer scales poorly. When training with Adam, we observed instability with default parameters (especially, epsilon). We changed it to 0.01 and 1.0 and have two runs which show dramatically different results. Our initial reason for not including comparisons to Adam was that we wanted to use standard models and training parameters (i.e., the Inception and Resnet papers use RMSprop).
>
> We think that the significance in our paper lies in the strong experimental results:
> 1. Significantly improved accuracy in output models (using a small number of workers) over published baselines -- i.e., just switching over to our optimizer will increase accuracy by 0.8-0.9%.
> 2. Excellent scaling behaviour (even using a very large number of workers).
> For example, our experimental results for (2) are strictly stronger than those in the literature for large batch training.
>
> The results that we have hold for ImageNet, but also for CIFAR-10 and CIFAR-100 with no change in hyperparameters, so we think that the results are likely to carry over to most modern CNN architectures on image datasets -- the hyperparameter choice will likely work out of the box (much like the beta_1, beta_2 and epsilon parameters in Adam). We agree that there appears to be quite a few hyperparameters, but \alpha and \beta are regularization coefficients, so they have to be roughly scaled to the loss; \gamma is a moving average coefficient and never needs to be changed; \mu is dependent only on time, not the model; finally, training is quite insensitive to K (as mentioned in Section 3.2). Thus, the only hyperparameter that needs to be specified is the learning rate \eta, and that does determine the speed of optimization.

---

### Public Comment · ~Boris_Ginsburg1 · 2017-10-31
**Excellent paper! Questions and comments ...**

Thanks very much for this outstanding paper! It is very interesting, both from the theoretical and practical points of view.
We have several questions:
1.	Page 5: The transition from Eq. 7 to eq. 9 intuitively makes sense, but we would appreciate more rigorous derivation (maybe as Appendix B?)
2.	Page 6: can you please clarify the transition from Alg. 1 to Alg. 2. ?
* Alg. 1 line 5 uses fixed w_t and fixed mini-batch.
* Alg 2, line 9-11 uses different w_t and different batches (similar to regular SGD).
What are the assumptions / requirements which make it possible to use different w_t?
3.	Page 6, Alg. 2: is periodic reset of m_t necessary?
What will be the impact on performance if we don’t reset m_t and avoid K at all?
4.	Page 6, Alg 2: Is Alg. 2 w/o regularization equivalent to regular SGD with adaptive momentum?
5.	Page 8, section 4.2: Did you try to use Neumann optimizer for training networks w/o batch norm (e.g. Alexnet or Googlenet) with large batch?
6.	Page 8, section 4.3 “Regularization”: Did you try to use L2-regularizer (weight decay) instead of “cubic + repulsive term”?

Typos:
1.	Page 3 , eq. 3: ‘-“ sign is missing
2.	Page 3 , after eq. 6: variable z is not defined (is z:= (w-w_t)/ \nu ? )
3.	Page 4 , first equation (line 4): should be O(|\nu m_t|^ 2 ?
4.	Page 5, eq.9: last term should be m_{k-1}
5.	Page 6, Alg. 2, line 11: should be w_t=w_{t-1} + m_t ?
6.	Page 6, Alg. 2, line 13:  should be return w_T?
7.	Page 7: can you set a right reference to data augmentation strategy?

Thanks again for an excellent paper!

---

> ### Author Response · Authors · 2018-01-05
> **Combined response to Boris Ginsburg's comments**
>
> Thanks for your interest and detailed feedback, Boris. We’ve incorporated most of your feedback, and hope to answer some of your questions below:
>
> 1. We’ve added the small calculation for this in Section 3.1.
>
> 2. A couple of things are going on here:
> i) We allow the mini-batch to vary in algorithm 2. This is a pretty significant change (we like to think of solving a stochastic bootstrap style subproblem instead of deterministic ones).
> ii) We change the notation to offset the w_t (so that w_t in Algorithm 2 actually correspond to w_t + \mu m_t in Algorithm 1). This is a pure notational change, and has no effect on the iteration -- we could also have done the same thing for Algorithm 1 (i.e., we could have unrolled m_t as the sum of gradients).
>
> 3. It seems somewhat insensitive to period of the resets, but the resets are necessary, especially at the start.
>
> 4. The coefficients we have for m_{t-1} and d_t aren’t a convex combination, and additionally, we subtract an extra \eta d_t from the update in Line 11 (this subtraction is correct, and somewhat surprising...you accidentally identified it as a typo below). It’s somewhat difficult to reinterpret as momentum.
>
> 5. We have not tried on large models without batch normalization. Since most convolutional architectures include batch norm, we had not thought to have experiments along this axis.
> 6. By default, all the models we used include weight decay -- so Figure 5 (the ablation experiments) should give you an idea of what happens if you use weight decay and not cubic + repulsive.
>
> Typos:
> We have all except (5) and (6) -- thank you! (5) and (6) are actually correct -- it definitely looks a little strange, but what we’ve done is to keep track of offset variables w_t + \mu m_t.

---

### Public Comment · (anonymous) · 2017-12-14
**Questions**

How many P100 did you need to use in order to fit so large batches?
Is the algorithm unstable in low batch size regimes?

---

> ### Author Response · Authors · 2018-01-05
> **Details on our experiments**
>
> On the Resnet-V1, each P100 had 32 examples, so 32000 corresponds to 1000 GPUs. This is updated in Table 3 now.
>
> To your question about small batches -- we ran the algorithm in asynchronous mode ( mini-batches of 32, with 50 workers doing separate mini-batches); the final output was quite a bit worse in terms of test accuracy (76.8% instead of 79.2%). It’s not clear whether it’s the Neumann algorithm with batch size 32 or the async that causes the degradation though.  So at the least algorithm doesn’t blow up with small batches, but we haven’t explored this setting enough to say anything conclusive.

---

### Author Response · Authors · 2018-01-05
**Changes made to the updated paper**

We have made the following changes in the new version of the paper that we uploaded.

We have added some new experiments

(1) Comparison to Adam in Figure 1,
(2) Multiple Initializations in Appendix D, and
(3) A Stochastic Convex Problem in Appendix B, along with small edits suggested by reviewers.

---

### Author Response · Authors · 2018-01-05
**Overall response to reviewer's comments**

We thank all the reviewers for their feedback. Before we address individual comments, we would like to mention some key themes in this paper that seem to have been lost mainly due to our presentation in the experiments section.

(1) Training deep nets fast (in wall time/parameter updates) without affecting validation performance is important. Previous attempts to scale up, using large batch size and parallelization, hit limits which we avoid. For example, using 500 workers computing gradients in parallel, we can train Resnet-V1-50 to 76.5% accuracy in a little less than 2 hours. In contrast, in Goyal et al. “Accurate, large minibatch SGD: Training Imagenet in 1 hour.”, the maximum batch size was 8000 (equivalent to 250 workers), and in You et al. “Scaling SGD batch size to 32k for imagenet training”, there is a substantial 0.4-0.7% degradation in final model performance.

(2) Our method actually achieves better validation performance (~1% better) compared to the published best performance on image models in multiple architectures.

---

### Decision · Program_Chairs · 2018-01-29
**ICLR 2018 Conference Acceptance Decision**

**Decision:**

Accept (Poster)

**Comment:**

Pros:
+ Clearly written paper.
+ Easily implemented algorithm that appears to have excellent scaling properties and can even improve on validation error in some cases.
+ Thorough evaluation against the state of the art.

Cons:
- No theoretical guarantees for the algorithm.

This paper belongs in ICLR if there is enough space.